# U-Net Convolutional Neural Network for Mapping Natural Vegetation and Forest Types from Landsat Imagery in Southeastern Australia

**DOI:** 10.3390/jimaging10060143

**Published:** 2024-06-13

**Authors:** Tony Boston, Albert Van Dijk, Richard Thackway

**Affiliations:** Fenner School of Environment and Society, Australian National University, Acton, ACT 2601, Australia; albert.vandijk@anu.edu.au (A.V.D.); richard.thackway@anu.edu.au (R.T.)

**Keywords:** vegetation mapping, convolutional neural networks, random forests, Landsat imagery, segmentation, spatial and temporal variation, deep learning, remote sensing

## Abstract

Accurate and comparable annual mapping is critical to understanding changing vegetation distribution and informing land use planning and management. A U-Net convolutional neural network (CNN) model was used to map natural vegetation and forest types based on annual Landsat geomedian reflectance composite images for a 500 km × 500 km study area in southeastern Australia. The CNN was developed using 2018 imagery. Label data were a ten-class natural vegetation and forest classification (i.e., Acacia, Callitris, Casuarina, Eucalyptus, Grassland, Mangrove, Melaleuca, Plantation, Rainforest and Non-Forest) derived by combining current best-available regional-scale maps of Australian forest types, natural vegetation and land use. The best CNN generated using six Landsat geomedian bands as input produced better results than a pixel-based random forest algorithm, with higher overall accuracy (OA) and weighted mean F1 score for all vegetation classes (93 vs. 87% in both cases) and a higher Kappa score (86 vs. 74%). The trained CNN was used to generate annual vegetation maps for 2000–2019 and evaluated for an independent test area of 100 km × 100 km using statistics describing accuracy regarding the label data and temporal stability. Seventy-six percent of pixels did not change over the 20 years (2000–2019), and year-on-year results were highly correlated (94–97% OA). The accuracy of the CNN model was further verified for the study area using 3456 independent vegetation survey plots where the species of interest had ≥ 50% crown cover. The CNN showed an 81% OA compared with the plot data. The model accuracy was also higher than the label data (76%), which suggests that imperfect training data may not be a major obstacle to CNN-based mapping. Applying the CNN to other regions would help to test the spatial transferability of these techniques and whether they can support the automated production of accurate and comparable annual maps of natural vegetation and forest types required for national reporting.

## 1. Introduction

Almost 91% of Australia is occupied by vegetation [1,2], with forest ecosystems covering about 17% of the continent [3] concentrated in higher rainfall coastal areas. Understanding the spatial and temporal occurrence of vegetation is important to scientists and natural resource managers to support nature conservation, ecosystem science and resource inventories, sustainable land use, forestry, and agriculture [4,5,6]. Modelling the distribution of species based on bioclimatic variables using a variety of algorithms has a long history in ecology, e.g., [7,8,9,10]. Mapping vegetation based on the interpretation of remote sensing imagery using deep neural networks is a more recent approach [11]. 

Field surveys provide high-quality, detailed information about vegetation distribution, trends, and conditions [12,13] but are time-consuming and expensive compared with remote sensing techniques. Remote sensing methods can be more repeatable, consistent, economical, and reliable than field mapping [14,15]. However, current image classification methods have limitations associated with the spatial, temporal, and spectral resolution of the imagery, require significant input from experts to perform well, and typically do not generalise well spatially or temporally without significant work [16].

Australian vegetation and land cover mapping using remote sensing imagery has progressed from resolutions of 250 m to 1 km using MODIS (moderate resolution imaging spectroradiometer), e.g., [17], to approaches using multi-temporal Landsat data at 30 m resolution, e.g., [18,19,20]. Studies using higher resolution imagery have been applied to small areas to successfully map Australian forest communities such as those dominated by *Acacia*, *Angophora*, *Callitris* and *Eucalyptus* [21], turpentine, rainforest and eucalypts [22] and *Allocasuarina*, *Angophora*, *Banksia* and *Eucalyptus* spp. [23], but multi-temporal high-resolution maps with continental coverage are currently not publicly available.

Machine learning methods that combine spectral with spatial or textural features may improve vegetation mapping results compared with pixel-based, spectral-only methods [24,25]. Specifically, convolutional neural networks (CNNs) combine the detection of spectral and spatial features and have been applied to land cover and vegetation mapping using optical data, e.g., [4,26,27,28,29,30]. These deep learning techniques have shown improvements over traditional machine learning methods such as random forests (RF), which generally produce good classification results with fast processing speeds and were arguably the leading approach before CNNs [31,32]. RFs are typically used as a benchmark against which to compare CNN results, e.g., [33,34]. However, in two reviews of the literature, the majority of CNN studies (88%) used higher resolution satellite data of 10 m or finer [11,35].

In contrast, this study was motivated to understand whether CNNs have merit for mapping natural vegetation and forest types using medium-resolution Landsat imagery, which would enable the use of the long time series of Landsat data to monitor environmental change. Specifically, the U-Net CNN was chosen for use in this study as reviews of the literature in 2022 suggested that U-Net was the most commonly used encoder/decoder CNN for segmentation of remote sensing images. This study did not aim to undertake a detailed comparison of deep learning model architectures but rather focussed on the application of the two leading methods identified from the literature (U-Net CNN and RF) to natural vegetation and forest type mapping in southeastern (SE) Australia, investigating the performance and features of maps generated by the models for different vegetation classes. Previous studies by the authors [36,37] have shown that label data of six classes representing broad land cover classes inferred from one-off, imperfect, and coarser mapping of land use can be used to train a U-Net CNN that performed notably better than a spectral-only RF method, and when independently verified, in fact, appeared more accurate than the label data.

Our goal in this study was to determine if similar promising results can be obtained when considering different vegetation classes. A priori, we expected that the lesser importance of clearly recognisable landscape features (e.g., waterways and infrastructure) and the potentially gradual transitions between vegetation types would make this mapping more challenging at Landsat resolution. Our specific objectives were as follows:Test if a U-Net CNN model using Landsat data can map natural vegetation and forest types and help to understand their change over time in SE Australia;Quantify the effectiveness of the method and its temporal stability;Compare CNN results with vegetation maps generated using RF.

Our approach was to build CNN models using 2018 annual Landsat geomedian data as input and label data compiled from imperfect reference land cover and native vegetation maps for a 500 km × 500 km study area of SE Australia. The model with the highest validation accuracy was tested on a 100 km × 100 km area south of the study area and compared with results obtained using RF. 

Annual Landsat geomedian data rather than time series composites were considered sufficient as input data for CNN modelling as Australian forest types in the study area are evergreen and typically do not exhibit seasonal phenological changes, having constant moderate to high enhanced vegetation index values with little temporal variability [38]. In addition, a previous study [36] showed that three measures of the annual temporal variance from the median in input Landsat data did not significantly improve the accuracy of the developed CNN models for this area.

The CNN model developed in this study was applied annually for the period of 2000–2019 for the test area, and the temporal stability of time series results was investigated. As a final step, to assess the accuracy of CNN model results with respect to an independent, high-quality reference, they were compared with vegetation survey plots for the study area.

## 2. Material and Methods

### 2.1. Input Data

Model development was undertaken using the 2018 annual Landsat 8 geomedian reflectance data at 25 m resolution [39] derived through pixel-based image compositing [40,41]. The annual composites are based on a statistical value, the geometric median, that preserves the spectral relationships between bands and improves consistency across the scene boundaries and spatial noise compared with alternative compositing methods [42]. The study area for the experiment was composed of 25 Landsat tiles in GDA94 projection (each ~100 km × 100 km) from NSW, ACT and Victoria, which were used for training and validation of models (large RGB image in Figure 1), along with a southern test area (tile +15, −42: yellow box in Figure 1).

The six bands (Blue, Green, Red, NIR, SWIR1, SWIR2) provided in the 2018 Landsat 8 geomedian data were combined as a uint16-encoded GeoTIFF of the study area. The GeoTIFF file was split into non-overlapping 128 × 128-pixel patches and converted to NumPy array format [43] as input for model building.

### 2.2. Label Data

‘Circa 2018’ label data were created based on the current best-available regional-scale continental coverages by compiling the 100 m resolution Forests of Australia (FoA) map [3], the 50 m Catchment scale Land Use of Australia (CLUM) map [44] and the 100 m National Vegetation Information System (NVIS) V6 map [1].

These three resources were downloaded and examined in detail using the Geographic Information System QGIS, allowing overlay, analysis, and comparison to high-resolution satellite imagery. The FoA map was simplified to a smaller number of classes to be used as label data (Table 1); see [45] for more detailed descriptions of major forest types, including images of species, structure, and form. One-to-one mapping was used for the following FoA classes: Non- Forest, Acacia, Casuarina, Callitris, Mangrove, Melaleuca and Rainforest. All FoA Eucalypt subclasses (including open and closed forest, mallee, woodland, low, medium, and tall forest) were amalgamated into a single Eucalyptus class. All planted forest subclasses (Hardwood, Softwood, Mixed Species Plantation, and the Other Forest subclasses representing planted forests) were mapped to a single Plantation class. The Other Native Forest class was mapped to Eucalyptus or Plantation by comparing it to the Plantation land use class in the CLUM map.

Label data were further validated and refined through comparison with NVIS, and high-resolution imagery was interpreted visually to develop heuristic rules to reconcile the mapping sources where necessary. For the study area, NVIS included slightly larger extents for Mangrove and Melaleuca than FoA, and these additional areas were added to these classes. Acacia was limited to areas where it occurred in both FoA and NVIS (NVIS classes: Acacia Forests and Woodlands, Acacia Open Woodlands, Acacia Shrublands) but extended with coastal Acacia (NVIS class Acacia Shrublands), which was not mapped in FoA. In addition, a Grassland class was included in the label data based on CLUM (classes 210: Grazing native vegetation, 320: Grazing modified pastures and 420: Grazing irrigated modified pastures). Finally, the resulting label data were resampled to 25 m to match Landsat data resolution. The sources of the label data are summarised in Table 1, and the resulting map is shown in Figure 2.

### 2.3. Modelling Methodology

A U-Net CNN [46] and RF model were created using input 2018 Landsat geomedian data for the study area with the ten-class vegetation classification (Figure 2 and Table 1) as label data. The CNN results were compared with those of the RF algorithm for a test tile south of the study area (Figure 1). 

For the CNN, developed using the segmentation modelling software [47], ResNet50 v1 [48] was used as the encoder for the U-Net model and transfer learning [49] was employed. The ResNet50 portion of the U-Net CNN model was initialised using weights from the ImageNet ILSVRC-2012-CLS dataset training for the RGB channels [50]. The remaining input bands (NIR, SWIR1, SWIR2) were trained from scratch. No layers were frozen, and all layers (encoder and decoder) were trained or fine-tuned during model building. Using a combination of contracting (dimension reductions of 128 × 128, 64 × 64, 32 × 32, 16 × 16, 8 × 8) and expanding paths with five layers and skip connections, the U-Net model was able to extract and capture features in an image, while also retaining and reconstructing spatial information to produce good segmentation maps.

The six-band 2018 annual Landsat 8 geomedian GeoTIFF image (19,968 × 19,968 pixels) representing the study area was split into 24,336 non-overlapping 128 × 128-pixel patches. The land-based patches (23,642) were split randomly into training and validation sets, with the remaining 694 ocean patches discarded. The test set was based on 955 land-based patches for the test tile 2018 annual Landsat 8 geomedian image (3968 × 3968 pixels) south of the study area (Figure 1 and Table 2).

A sensitivity analysis was undertaken to determine the optimal parameters for building a neural network model, producing the Adam optimiser [51] with a batch size of eight and a learning rate of 0.0001. Dice and focal loss (after [52]) were used to address class imbalance:$${Loss}_{total}={Loss}_{Dice}+(1\times {Loss}_{Focal})$$

Data augmentation is a powerful technique to increase the data available for model building and help prevent model overfitting. The fast augmentation library ‘Albumentations’ [53] was used for data augmentation, applying transformations to modify patches, including Gaussian noise addition applied for each channel independently, image blurring using a (3, 3) kernel and sharpening, flipping the input horizontally around the y-axis, random four point perspective transforms, random cropping and affine transformations to translate, scale and rotate the input, applied dynamically during training to the 128 × 128-pixel patches. CNN models were built for up to 80 epochs in iterations of ten epochs. Models with the highest validation accuracy were run against the test tile (Figure 1) to evaluate their performance compared with label data and checked visually. After ten epochs, if validation accuracy failed to improve, the Keras callback ReduceLROnPlateau [54] was used to reduce the learning rate.

RF modelling was undertaken using the Python scikit-learn package [55]. The number of trees/estimators (50, 100, 200, 400, 600, 800, 1000) and maximum depth (10, 20, 30, 40) were varied to find the optimal test results using balanced class weights inversely proportional to class frequencies in the input data. All combinations of the number of estimators and tree depth (7 × 4 = 28 combinations) were tested, with results examined quantitatively using confusion matrix metrics and qualitatively through visual inspection to find the best combination. To remain within memory limitations, models were created based on study area subsets of 0.2% (~800,000) pixels, selected using stratified random sampling proportional to the area occupied by each label class as input to the RF classifier.

RF modelling is generally less resource-intensive than models built using CNNs. In this study, models built for 80 epochs using the U-Net CNN took approximately eight hours on a Google Colab Pro single GPU machine, while model builds for random forest on a single CPU were completed in less than ten minutes.

CNN and RF modelling results were assessed using statistical measures [56] derived from confusion matrices, which are commonly used to analyse the effectiveness of categorical classifications to understand the accuracy of generated vegetation maps from different perspectives. Metrics included overall accuracy (OA), producer’s accuracy (PA), user’s accuracy (UA) and F1 scores by class, as well as the weighted-mean F1 (with weights based on class abundance) and Kappa [57] scores, as described by [36].

The CNN with the highest performance was applied to the test tile (Figure 1) for 2000–2019 using Landsat 7 (2000–2012) or Landsat 8 (2013–2019) annual geomedian reflectance as input. The temporal stability of the CNN results for these 20 years was examined in detail following two approaches:Year-on-year changes for all years (2000–2001, 2001–2002, …, 2018–2019) were compared using confusion matrices to determine the similarity of model results for each of the 19 annual transitions.The total number of CNN class changes for all 16 M pixels in the test tile for the 20 years was determined. These results were further subdivided by vegetation class, using the 2018 label data for the tile as the reference to examine the temporal stability of each predicted model class.

Multiple temporal vegetation transitions can be realistic, especially for some classes (e.g., plantation forest), but frequent class changes were considered more likely to indicate mapping uncertainty.

To independently verify the validity of label data and CNN results, vegetation survey plots for each vegetation class were extracted from the CSIRO HAVPlot dataset [58], a compilation of over 200,000 Australian vegetation survey plots and approximately 6 million observations of the occurrence of flora, predominantly recorded at the species level. HAVPlot species observations for each vegetation class were compared with 2018 label and model data for the study area, and the number of matches was determined. Plot location data and associated observations of plants and their crown cover percentage (also known as canopy closure, canopy cover, vertical canopy cover or crown projective cover [59]) were first extracted from HAVPlot for the study area. Plot observation dates ranged from 2015 to 2021 (plus or minus three years from our target year of ‘circa 2018’), but only about 20% of HAVPlot observations in the study area had an associated date. Plots were excluded where they occurred within the Non-Forest class (38% of the total study area) of the label data (Figure 2).

For forest classes, plots of observations with a crown cover ≥20% were included, matching the Australian definition of a forest used by the National Forest Inventory [60]. This criterion (cover ≥20%) was also used for the Grassland class. In addition, plots were excluded where the difference between the highest abundance (crown cover) and second-highest abundance was less than 20%, yielding a total of 6467 plots for the study area (Figure 3).

Broad vegetation types were extracted from HAVPlot survey sites using the criteria for inclusion in Table 3, with the number of plots indicated. There were insufficient (six) Mangrove plots from the HAVPlot dataset in the study area. An additional 27 ‘plots’ (observations of occurrence of *Avicennia marina*) were added from the Atlas of Living Australia (www.ala.org.au accessed on 1 June 2023). The Melaleuca plots excluded *Melaleuca uncinata*, commonly known as brushwood, broom bush or broom honey myrtle, as it occurs predominantly to the west of the study area and is not represented in the Melaleuca class of the label data compiled from FoA or NVIS V6 (Table 1 and Figure 2).

The HAVPlot dataset focuses on natural vegetation and generally does not include plots dominated by *Pinus radiata* or other production trees. To include this class, 200 ‘virtual plots’ were created by randomly selecting pixels from the study area found within the 2018 raster coverage of softwood plantations in SE Australia. Through visual inspection of high-resolution imagery from 2018, this set was further reduced to 190 ‘plots’ that could be confirmed to represent plantations.

The resulting 6467 vegetation survey plots were used and, in a second experiment, further subdivided into subsamples using different criteria to investigate possible sources of bias in validation as follows:All plots where only one species of interest (Table 3) was observed (4620 plots);Plots with ≥50% crown cover (3456 plots);Plots of ≥400 m^2^ area − generally as 20 m × 20 m (5439 plots);Plots with one species of interest and ≥50% crown cover (1667 plots);Plots with one species of interest and ≥400 m^2^ (3785 plots);Plots with ≥50% crown cover and ≥400 m^2^ (3043 plots);Plots with one species of interest, ≥50% crown cover and ≥400 m^2^ (1436 plots).

Each combination of these criteria was evaluated, and statistics were generated to evaluate the accuracy of label data and the CNN predictions. Note that for the species where a range of percent crown cover values were listed in HAVPlot, the upper value was used.

Finally, to assist in interpreting differences in performance, reflectance in all bands was extracted for the study area from the 2018 annual Landsat geomedian data for each vegetation class. This was performed to test if distinct spectral properties explain the RF performance and may contribute to the CNN’s ability to distinguish each class, in addition to textural or spatial features.

## 3. Results

U-Net CNN results, label data and 2018 Landsat 8 geomedian input are shown for the study area in Figure 4. Figure 5 shows similar results for the test tile south of the study area (Figure 5a–c) and RF results (Figure 5d).

The test tile is dominated by Eucalyptus along with Plantation forests and Grassland, with smaller areas of Rainforest, Casuarina, Melaleuca, Acacia and coastal Mangrove classes. The CNN had an overall accuracy of 93%, weighted-mean F1 for eight classes of 93% and Kappa of 86% (Table 4). The Callitris class was not found in the test tile, as it mainly occurs in dryer regions further west. The RF model had a lower overall accuracy of 87%, weighted-mean F1 for eight classes of 87% and Kappa of 74% (Table 5). The RF showed isolated pixel issues (‘salt and pepper’), especially for the Plantation class, and lower performance, particularly for minor classes.

Metrics calculated from the confusion matrices (Table 4 and Table 5) show the higher weighted mean F1, Kappa and OA results for the CNN and large differences between the CNN and RF per class F1 scores for less common classes, such as Plantation, Mangrove, Rainforest and Acacia (Figure 6).

Examples of CNN results highlight its utility in detecting different vegetation classes. In the first row of Figure 7, the CNN (Figure 7c) shows a reasonable match to label data (Figure 7b) for the coastal Acacia class and Mangrove and Rainforest. The Eucalyptus and Grassland classes are generally very well detected by the CNN. Casuarina and Melaleuca are generally less well-detected or undetected when compared with label data. In the second row (Figure 7f), Callitris is well detected with a complex pattern of occurrence. Instances of the Acacia class away from the coast are generally either undetected or much less accurately detected by the CNN.

In the third row (Figure 7i), the model reasonably detects all classes. Casuarina commonly occurs along water courses, e.g., the river she-oak (*Casuarina cunninghamiana*), and in this instance, it is reasonably well detected by the CNN, except for some areas in the southeast of the image. The Rainforest and Plantation classes are also well detected, with a good match between the model result (Figure 7i) and label data (Figure 7h).

### 3.1. Temporal Stability of CNN

The CNN was applied to the test tile for the years 2000–2019. Changes over these 20 years (Figure 8 and Table 6) were analysed to understand the temporal stability of the CNN-based mapping.

Nineteen one-year transitions (i.e., 2000–2001, 2001–2002, …, 2018–2019) were compared, and confusion matrices were calculated for each transition. Overall accuracy of these one-year transitions varied from 94 to 97%, weighted mean F1 from 95 to 97% and Kappa from 89 to 93%. 

An understanding of temporal stability can be derived by calculating how often pixels change vegetation class over consecutive years (2000–2019). For the 20 years, a pixel class change every year would yield 19 changes. For the test tile, the vegetation class of 76% of pixels did not change over the 20 years (white area in Figure 9a). Of the remaining 24% of pixels that did change, two-thirds (16% of the total area) showed only 1–4 changes (Figure 9b).

A further analysis was carried out to examine whether the temporal instability of particular vegetation classes could indicate uncertainty in the mapping of that class. Using the 2018 label data as a reference, each class of the test tile was analysed for the period 2000–2019 to determine the number of changes by class (Figure 10).

The Eucalyptus, Grassland, Non-Forest, and All classes curves (Figure 10) were similar, with about 75–80% of pixels showing no changes over 20 years, whereas the other classes had fewer pixels without change.

### 3.2. Comparison of CNN Results to HAVPlot Vegetation Survey Data

An independent accuracy assessment was undertaken using the HAVPlot vegetation survey plot data [58]. Various criteria were used to subset the plot data with the best results obtained for plots with ≥50% crown cover (3456) where there were 2745 matches (79%) of the plot class with the CNN class and 2544 matches (74%) of the plot class with the label class (Appendix A, Table A1).

For the vegetation survey plots with ≥50% crown cover, confusion matrices comparing the CNN model with plot data and label data with plot data were generated. Overall accuracy was 81% and 76% for model and label data vs. plot data, with 85% and 78% weighted-mean F1 scores and Kappa of 66% and 55% (Figure 11).

Table A2 (Appendix A) provides spectral band statistics from the 2018 annual Landsat geomedian data of each vegetation class generated by the CNN model for the study area. For each vegetation class, the minimum, maximum, mean and standard deviation for each band are listed. Band ratios such as Green/Red (G/R), as well as indices such as NDVI and NDWI, were also calculated for each vegetation class. The spectrally diverse Non-Forest class (a mixture of crop and horticulture, water, urban and ocean) is not included in the table but does contribute to the ‘All Classes’ statistic, which represents the spectral properties and variation of the entire study area (~400 M pixels).

## 4. Discussion

At the broad level (Figure 4), the CNN prediction matched well with the label data. For the test tile, the correspondence between the CNN result and label data was very good (Figure 5, Table 4), with an overall accuracy of 93%. However, this tile has a highly imbalanced class distribution dominated by Eucalyptus (61% of pixels) and Grassland (28%). The relatively high overall accuracy and other metrics of both the CNN and, to a lesser extent, the RF model (Table 5) reflect the dominance of these classes (95% F1 scores of these classes for the CNN and 92% and 88% F1, respectively, for the RF model) and the relative ease of discriminating these classes in remote sensing imagery.

A closer examination of Table 4 and Table 5 assists in understanding the relative abilities of the CNN and RF methods to detect the less common classes such as Acacia, Casuarina, Mangrove, Melaleuca, Plantation and Rainforest. For these classes, the CNN produced significantly higher per-class F1 scores than the RF model (Figure 6). Plantation was much more effectively detected by the CNN than by the RF model (F1 was 87% vs. 47%), while Mangrove (49% vs. 10%) and Rainforest (22% vs. 5%) were detected by the CNN more than twice as effectively as by the RF. Acacia was quite well detected by the CNN in coastal locations (e.g., Figure 7c) but poorly where it occurs inland, and overall, was largely undetected by the RF model (15% vs. 0.4%). The location of coastal Acacia, commonly occurring on beach foredunes with high contrast to surrounding forest, beach, and ocean landforms, presumably aided its effective detection by the CNN using a combination of spectral and spatial/textural features. Even for classes that often cannot be detected effectively, the CNN results were better than for RF compared with the label data (e.g., Melaleuca: F1 4% vs. 0.1%; Casuarina 2% vs. 0.4%). The Callitris class does not occur in the test tile, but within the study area, it was detected (e.g., Figure 7f) with an efficacy similar to Mangrove (F1 55%).

From 2000 to 2019, the CNN results suggest a distinct trend of increasing Plantation and decreasing Grassland with a small increase in Eucalyptus forests (Figure 8 and Table 6). Table 6 shows an apparent increase in Plantation (38% over 2000 area) and a decrease in Grassland (6%), with a small increase in Eucalyptus forests (1.3%). This vegetation change is plausible and consistent with the recovery of part of the pre-European extent of the former Eucalyptus forest that had previously been cleared for pasture [61]. Changes in the extent of other classes that were less well detected by the CNN should be treated with caution, as the performance metrics suggest they are likely to be less reliable.

Although some year-to-year variation in class distributions was observed in CNN results over the 20 years (Figure 8c), one-year transitions (Figure 9) exhibited quite a high correlation, with OA between 94 and 97% and Kappa of 89–93%. The highest OA occurred for the dry years 2018–2019, one of which (2018) is the year the model was trained on, while the lowest OA was for the wet years 2011–2012, part of one of the strongest La Niña events on record from 2010 to 2012 [62].

Examination of the number of changes for 2000–2019 (Figure 9) and comparison to CNN results (Figure 8) shows many changes associated with particular vegetation classes such as Plantation and Rainforest. Other visible changes are associated with boundaries between classes, e.g., between Eucalyptus and Grassland, some areas of Non-Forest in the northwest predicted by the CNN for certain years and changes associated with coastal development. This study hypothesised that the degree of change by class might be a proxy for uncertainty in the mapping of that class (Figure 10). This broadly appears to be the case, as the best-detected classes (Eucalyptus, Grassland and Non-Forest) were the best conserved between years and showed the least number of changes. Among the other classes, Casuarina and Acacia appear more stable than expected when compared with Rainforest and Mangrove, which the CNN was better able to detect. The Plantation class displays a curve contrasting with the other classes. Plantation in the test tile increased by about 38% from 2000 to 2019 (Table 6) and was generally detected well by the CNN (e.g., F1 87%, Table 4), yet only about 22% of Plantation pixels were unchanged over this period (Figure 10). This contrasting result is most likely due to plantation management: when a plantation is first established or replanted, seedlings are generally not detected by the CNN. The number of years of growth required before seedlings are detected as Plantation is variable and may depend on various factors (e.g., the type of modelling, resolution and other characteristics of remote sensing imagery, the prevailing climatic conditions and the site condition including slope, aspect, position of coupe etc.). For the CNN, it appears that new or replanted plantation was alternately detected or not detected over about eight years as Plantation (see Figure 9a), resulting in the cross-cutting curve in Figure 10, which overlays the well-detected and temporally stable Eucalyptus and Grassland classes beyond that period.

Based on a comparison to the HAVPlot vegetation survey data for the study area, the CNN appears to be about 5% more accurate (OA) with higher weighted mean F1 and Kappa than the label data (Figure 11), suggesting that imperfect training data may not be a major obstacle to CNN-based mapping. A previous study found a similar result [36]. Class F1 scores of the CNN vs. HAVPlot data are higher or equal to the label data comparison for all classes except the poorly determined Acacia and Melaleuca classes. Class F1 scores for the model compared with plot data ranged from 97% for Plantation, 88% for Eucalyptus, 87% for Grassland, 78% for Mangrove, 73% for Rainforest, 52% for Callitris, 29% for Casuarina, 13% for Acacia, and 8% for Melaleuca, with similar relative accuracies for different vegetation classes (Figure 12f) as derived from comparison to label data (*R*^2^ = 0.81).

A close examination of the vegetation survey plot data, CNN model results, and high-resolution imagery suggest that the CNN detected a vegetation type more successfully where the percentage crown cover was higher (Figure 12a–e). For several classes, the number of matches of the CNN to plot data (producer’s accuracy) was positively correlated with the number of included plots based on percent crown cover: Callitris (*R*^2^ = 0.77), Grassland (*R*^2^ = 0.85), Rainforest (*R*^2^ = 0.84), and Casuarina (*R*^2^ = 0.92). For the Eucalyptus class, the number of matches to plot data was inversely correlated (*R*^2^ = 0.94). This may be due to increasing confusion between Eucalyptus and Rainforest as the crown cover increases to nearly 100%, with denser forests more likely to occur in high rainfall environments.

The overall accuracy calculated for the CNN vs. plot data of 81% was somewhat lower than expected (Figure 11). The plot survey sites include the location of the plot centroid in geographical coordinates. About 77% of plots extracted for the study area are square, while 84% are at least 400 m^2^ (20 m × 20 m for square plots), which is close to the resolution of our imagery (25 m × 25 m). Some mismatches between the CNN and HAVPlot may be due to image registration errors rather than issues with the CNN itself. Another probable error source is vegetation change between the plot survey date and our target year, 2018. It is not possible to effectively screen the HAVPlot data for this source of error as only about 20% of plots included the date of observation in the metadata.

Regarding class spectral data (Appendix A, Table A2), the Grassland class had the highest maximum and mean reflectance in all bands due to the sometimes-sparse cover of this vegetation type and the influence of dry season senescence on the annual geomedian reflectance. Rainforest showed the lowest visible reflectance values, much lower and quite distinct from Eucalyptus forest, within which most rainforest is embedded (e.g., in wet valleys). The Rainforest class also showed the highest Green/Red ratio and normalised difference vegetation index (NDVI) [63]. Its dense green canopy appears readily distinguished spectrally from the Eucalyptus forest, as also observed by [22]. Mangrove and Melaleuca (often located close to each other in coastal areas) had low maximum values in all reflectance bands. The Mangrove class showed much higher mean and standard deviations in visible bands and lower values in the infrared bands compared with Melaleuca. Positive values for the normalised difference water index (NDWI) [64] and negative values for NDVI for the Mangrove class suggest that a water background may play a role in its discrimination by the CNN. We conclude that spectral characteristics are likely to play a role in detection by the CNN, in addition to spatial or textural features for these classes.

There are several limitations of this study. It did not explore different combinations of Landsat bands or vegetation indices as input to modelling but assumed that six bands were optimal based on an earlier study by the authors [36]. That study was focused on broad-scale land cover mapping rather than the delineation of specific natural vegetation and forest types. In addition, different methods, such as K-fold cross-validation, were not applied to gain a better understanding of how robust results were to changes in dataset splits (Table 2) and the degree of variance under different dataset partitioning.

## 5. Conclusions

A CNN was used to map natural vegetation and forest types in SE Australia and their change over time. The CNN produced higher accuracy results (93% OA when compared with label data) than equivalent mapping using the pixel-based RF algorithm (87% OA) for a simple ten-class vegetation classification. The U-Net CNN was temporally stable with 76% of 16 M pixels in the 100 km × 100 km test area, not changing class over 20 years (2000–2019), with high stability between consecutive years of between 94 and 97% OA.

The ability of the CNN to detect the highly imbalanced vegetation classes in the study and test areas was variable, from relatively high accuracy for Eucalyptus, Grassland and Plantation to moderate accuracy for Callitris, Mangrove and Rainforest and low accuracy for Acacia, Casuarina and Melaleuca, when compared with the label data. Independent verification of the CNN results for the 500 km × 500 km study area against vegetation survey plots yielded an overall accuracy of the model of 81%, with similar accuracies for different vegetation classes as those derived through comparison with label data (*R*^2^ = 0.81, Figure 12f). Examination of the spectral properties of each vegetation class predicted by the CNN for the study area suggests that spectral properties (rather than spatial context only) may be particularly important in detecting the Grassland, Rainforest, and Mangrove classes.

Research directions worthy of future investigation include the application of transformers either in combination with CNNs or separately to such mapping (e.g., vision transformers (ViTs), [65]), which have shown good results with satellite imagery. Recent work from the biomedical imaging domain that combines the Mamba state space model (SSM) with U-Net CNN [66] or ViT [67] may also be a useful area for further work applied to the segmentation of remote sensing data.

We conclude that the U-Net CNN, built using input annual Landsat geomedian data, shows greater skill than pixel-based (i.e., spectral) methods in mapping natural vegetation and forest types in SE Australia and their change over time. Several innovative techniques to analyse a time series of model results were developed as a contribution to a multi-year analysis of how vegetation changes through space and time. Results were verified through comparison with high-quality reference data from independently collected vegetation survey plots to ensure robustness. For the classes detected by the CNN with moderate to high accuracies, further testing on transferability to other areas of Australia is required to determine the potential contribution of this technique to the production of reliable continental vegetation maps over the more than thirty-year period that Landsat data are available.

## Figures and Tables

**Figure 1 jimaging-10-00143-f001:**
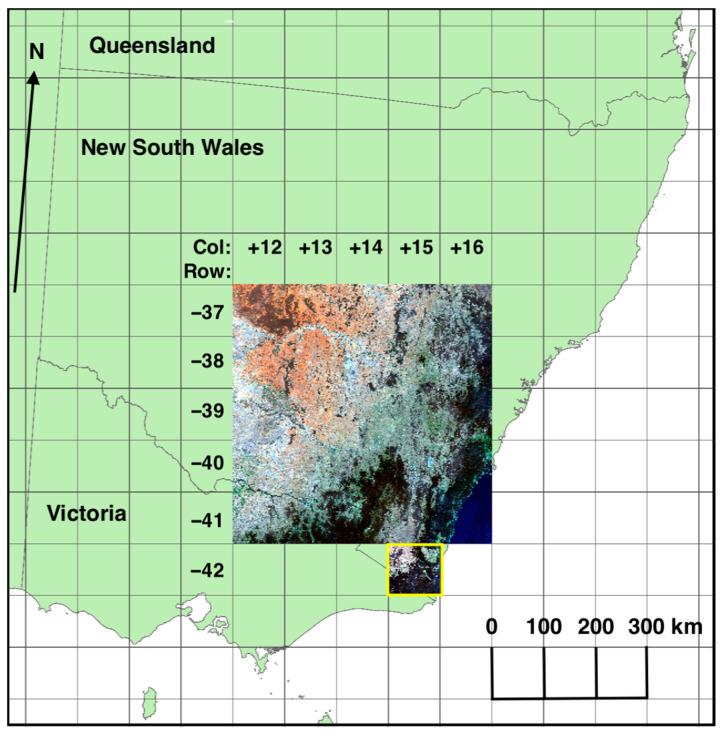
Study area (2018 Landsat 8 geomedian as true colour RGB), test tile +15, –42 (yellow box). Projection: GDA94/Australian Albers (EPSG: 3577).

**Figure 2 jimaging-10-00143-f002:**
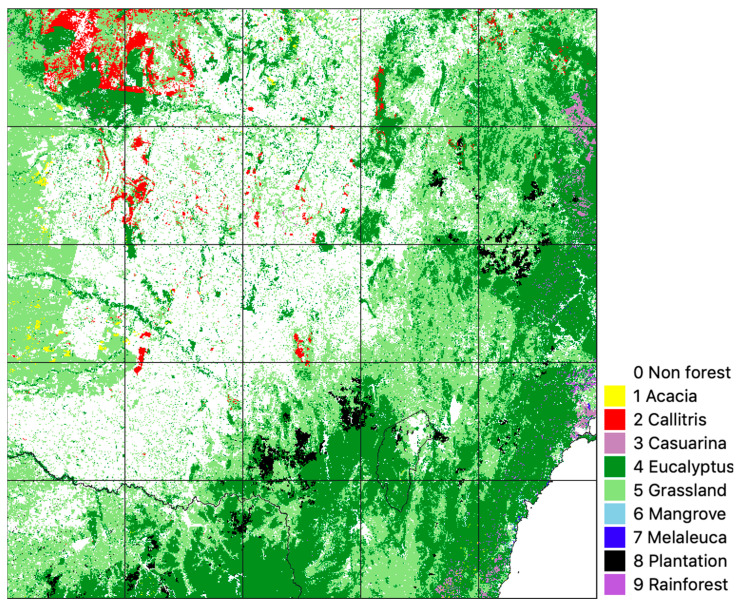
Label data (‘circa 2018’)—Natural vegetation and forest types for the study area (~500 km × 500 km). Projection: GDA94/Australian Albers (EPSG: 3577).

**Figure 3 jimaging-10-00143-f003:**
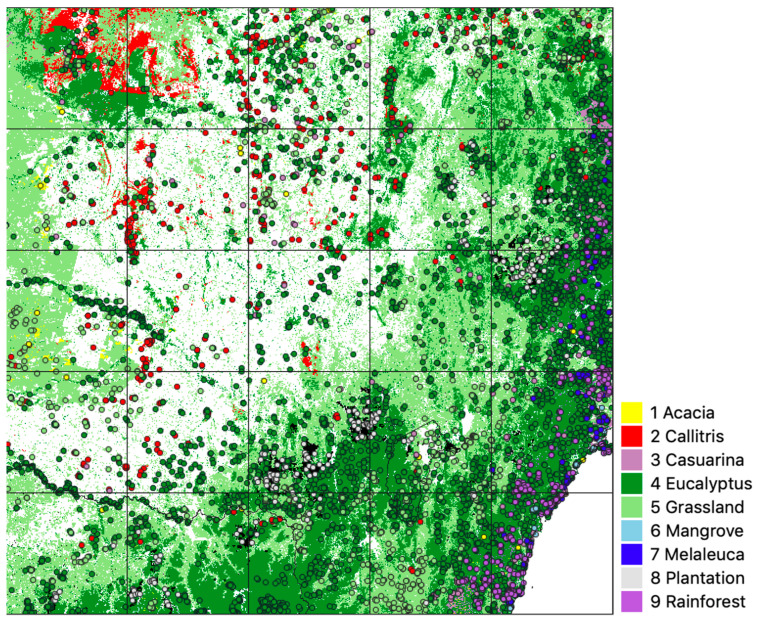
Location of 6467 vegetation survey plots for the study area displayed over label data. Plots (legend at right) were used to evaluate CNN and label data accuracy (base map legend as for Figure 2).

**Figure 4 jimaging-10-00143-f004:**
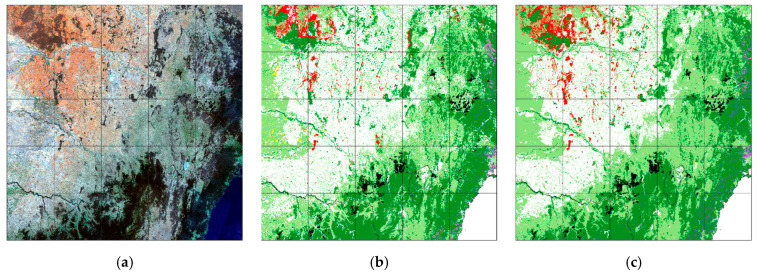
CNN (**c**) compared with ‘circa 2018’ label data (**b**) and 2018 Landsat geomedian (**a**) as true colour RGB for the study area (~500 km × 500 km) (map legend as for Figure 2).

**Figure 5 jimaging-10-00143-f005:**
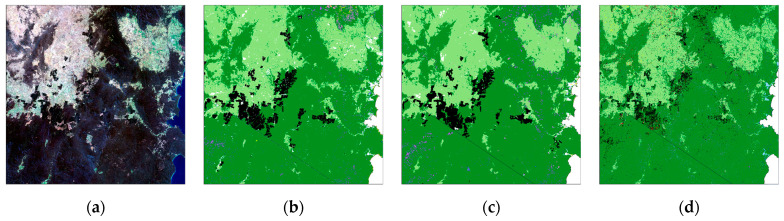
CNN (**c**) and random forests model (**d**) compared with ‘circa 2018’ label data (**b**) and 2018 Landsat geomedian (**a**) as true colour RGB for test tile (~100 km × 100 km). See Figure 2 for the map legend and Figure 1 for the geographic location of the test tile relative to the study area shown in Figure 4.

**Figure 6 jimaging-10-00143-f006:**
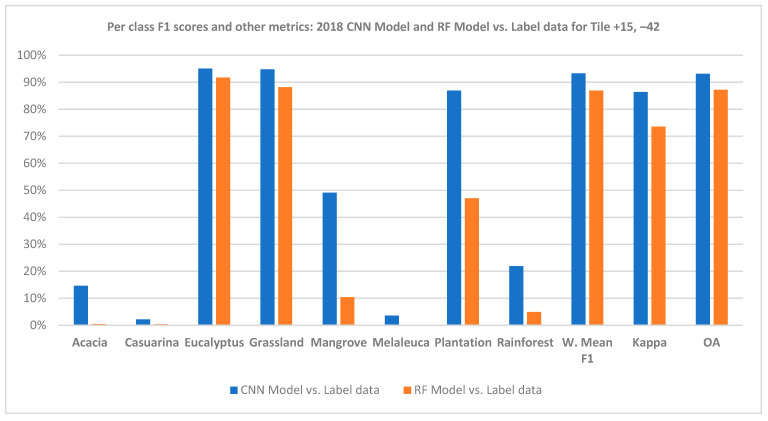
Class F1 scores, weighted mean F1, Kappa and OA for 2018 CNN and RF models vs. label data for the test tile. See Table 4 and Table 5 for source data. NB: Callitris does not occur in this tile.

**Figure 7 jimaging-10-00143-f007:**
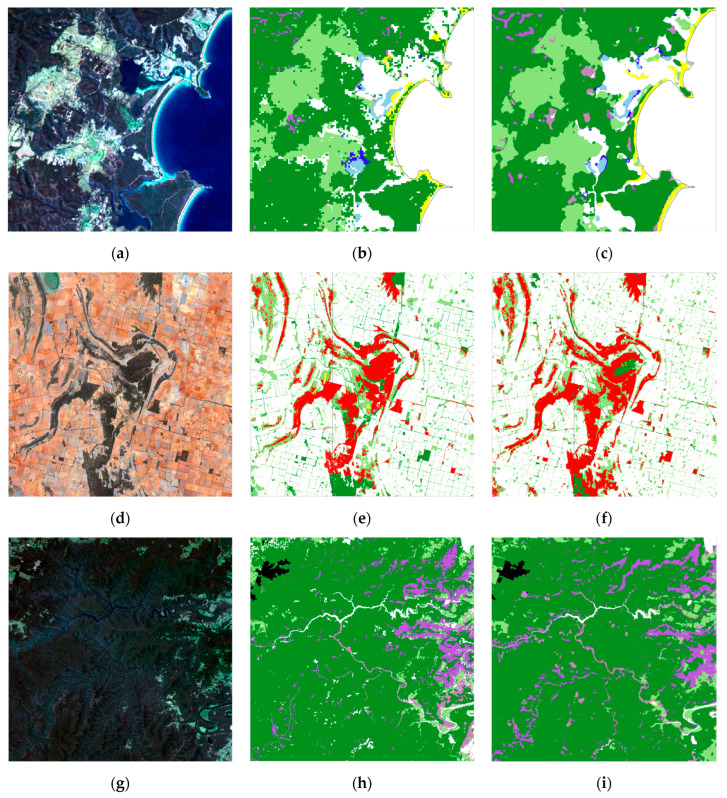
2018 Landsat geomedian as true colour RGB (first column), label data (second column) and CNN result (third column). Image dimensions and centroids: Test tile: (**a**–**c**) ~13 km × 13 km, 36.92° S, 149.88° E; Study area: (**d**–**f**) ~66 km × 66 km, 33.76° S, 146.22° E; (**g**–**i**) ~33 km × 33 km, 34.80°S, 150.36° E (map legend as for Figure 2).

**Figure 8 jimaging-10-00143-f008:**
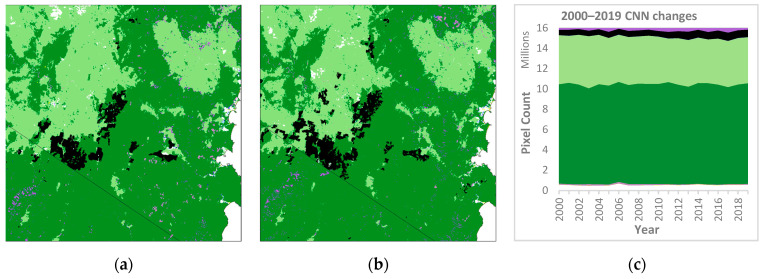
CNN results for 2000 (**a**) and 2019 (**b**) and vegetation class distribution (**c**) (legend as for Figure 2).

**Figure 9 jimaging-10-00143-f009:**
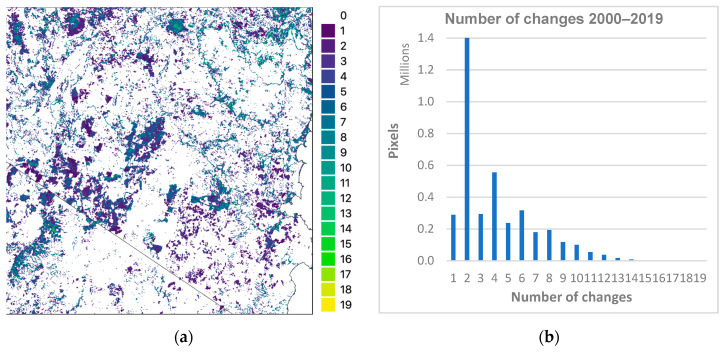
Pixel changes for 2000–2019 of CNN for test tile (**a**) and graph (**b**) showing distribution of changes.

**Figure 10 jimaging-10-00143-f010:**
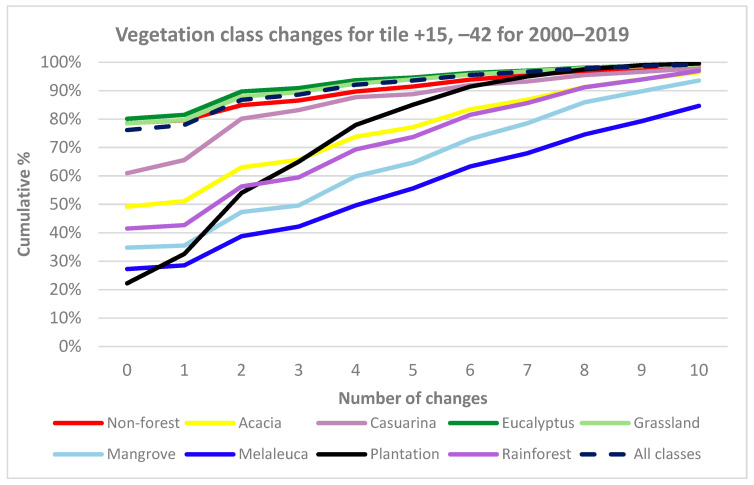
Plot of vegetation class changes for the test tile for 2000–2019 compared with 2018 label data.

**Figure 11 jimaging-10-00143-f011:**
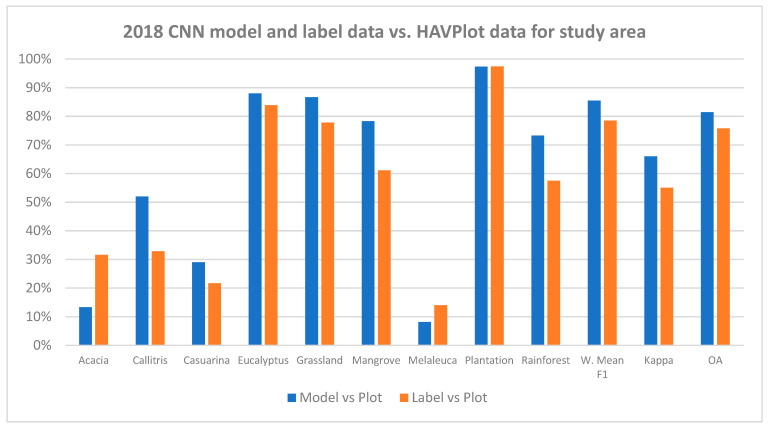
Class F1 scores, Weighted Mean F1, Kappa and OA for model and label vs. plot data.

**Figure 12 jimaging-10-00143-f012:**
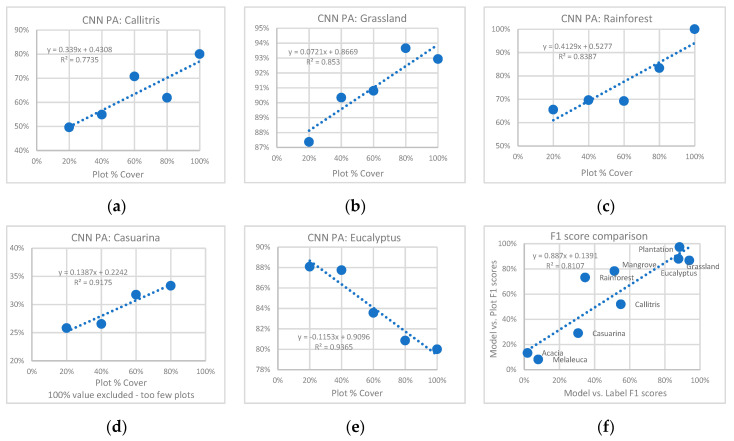
Correlation between CNN Model producer’s accuracy (PA) for the study area and plot percent cover for Callitris (**a**), Grassland (**b**), Rainforest (**c**), Casuarina (**d**) and Eucalyptus (**e**). Model vs. Label F1 scores plotted against Model vs. Plot F1 scores (**f**).

**Table 1 jimaging-10-00143-t001:** Mapping of source reference datasets to label data classes for the study area.

Class	Source	Comments
0 Non-forest	FoA	Based on FoA Non-Forest class. Non-Forest comprises crop and horticulture, water, urban and ocean. This class was reduced in area with the addition of the Grassland class from CLUM
1 Acacia	FoA/NVIS V6	Class is Acacia IF FoA AND NVIS V6 is Acacia plus coastal Acacia added from NVIS V6
2 Callitris	FoA	Better definition of Callitris forest vs. Non-Forest in NW of the study area in FoA compared with NVIS V6
3 Casuarina	FoA	In NE of the study area, a large extent of Casuarina in FoA is labelled Heathland in NVIS V6. Western Casuarina in NVIS V6 is not represented in FoA and does not look like forest in high-resolution satellite imagery. Southern areas of Casuarina are similar in both coverages. Jervis Bay Casuarina in FoA is labelled Eucalypt woodland in NVIS V6
4 Eucalyptus	FoA	All FoA Eucalypt and Eucalypt Mallee classes were combined. NVIS V6 Eucalypt classes are slightly more extensive in the west of the study area, but FoA appears to be more accurate when compared with high-resolution imagery
5 Grassland	CLUM	Grassland class based on CLUM, which is much more extensive than Grassland classes in NVIS V6 (19 Tussock Grasslands, 20 Hummock Grasslands and 21 Other Grasslands)
6 Mangrove	FoA/NVIS V6	Coverages are very similar. NVIS V6 has a slightly larger extent. Limited to coastal areas. Class is Mangrove IF FoA OR NVIS V6 is Mangrove
7 Melaleuca	FoA/NVIS V6	Coverages are very similar. NVIS V6 is slightly larger extent. Mainly limited to coastal areas. Class is Melaleuca IF FoA OR NVIS V6 is Melaleuca
8 Plantation	FoA	Good definition for softwood plantations. Very small areas on hardwood and mixed species plantation in northern Victoria. Plantation label class includes softwood, hardwood and mixed species plantations
9 Rainforest	FoA	NVIS V6 has no Rainforest in the study area. FoA coverage is largely limited to coastal ranges

**Table 2 jimaging-10-00143-t002:** Dataset splits for CNN modelling.

Dataset Splits	Count	Percent
Train	19,906	81%
Validation	3736	15%
Test	955	4%
TOTAL	24,597	100%

**Table 3 jimaging-10-00143-t003:** Criteria for inclusion and number of study area HAVPlot sites for each vegetation type.

Plot Veg. Type	Criteria for Inclusion	No.
Acacia	Genus = *Acacia* and no other major forest species present	78
Callitris	Genus = *Callitris*	536
Casuarina	Genus = *Allocasuarina* or *Casuarina*	344
Eucalyptus	Genus = *Eucalyptus* or *Corymbia*	4097
Grassland	Genus = *Austrostipa* or *Rytidosperma* or Species = *Aotus ericoides* or *Aristida ramose* or *Austrostipa bigeniculata* or *Bothroichloa macra* or *Carex appressa* or *Chrysocephalum apiculatum* or *Cynodon dactylon* or *Ficinia nodosa* or *Hemarthria uncinate* or *Juncus homalocaulis* or *Lomandra filiformis* or *Microlaena stipoides* or *Micromyrtus ciliate* or *Phragmites australis* or *Poa labillardierei* or *Poa poiformis* or *Poa sieberiana* or *Pteridium esculentum* or *Rumex brownie* or *Themeda australis* or *Themeda triandra* or *Typha domingensis* or *Viola banksia* or *Zoysia macrantha* and no major forest species present	768
Mangrove	Species = *Aegiceras corniculatum* or *Avicennia marina*	33
Melaleuca	Genus = *Melaleuca* and Species ≠ *Melaleuca uncinata*	76
Plantation	Species = *Pinus radiata*	190
Rainforest	Species = *Doryphora sassafras* or *Pittosporum undulatum* or *Syzygium smithii*	345
	TOTAL	6467

**Table 4 jimaging-10-00143-t004:** CNN vs. label data confusion matrix for test tile (excluding Non-Forest class). OA, weighted mean F1 = 93%, Kappa = 86%. Diagonal elements and totals in bold.

Pixel Count	Label Data									User’s
CNN	Acacia	Casuarina	Eucalyptus	Grassland	Mangrove	Melaleuca	Plantation	Rainforest	Total	acc. %
Acacia	**3396**	14	3531	1205	179	39	0	15	**8379**	40.5
Casuarina	2235	**1474**	49,844	10,490	3	77	213	160	**64,496**	2.3
Eucalyptus	26,315	60,837	**9,228,705**	236,205	768	797	106,197	85,648	**9,745,472**	94.7
Grassland	3385	2588	164,483	**4,189,872**	82	277	30,994	130	**4,391,811**	95.4
Mangrove	35	25	2292	135	**1960**	189	15	0	**4651**	42.1
Melaleuca	140	77	4547	198	336	**125**	49	0	**5472**	2.3
Plantation	0	0	55,363	9013	0	0	**669,958**	83	**734,417**	91.2
Rainforest	2607	4548	169,976	4659	0	0	233	**37,502**	**219,525**	17.1
Total	**38,113**	**69,563**	**9,678,741**	**4,451,777**	**3328**	**1504**	**807,659**	**123,538**	**15,174,223**	
Prod.’s acc. %	8.9	2.1	95.4	94.1	58.9	8.3	83.0	30.36		
F1 %	14.6	2.2	95.0	94.8	49.1	3.6	86.9	21.9		

**Table 5 jimaging-10-00143-t005:** RF vs. label data confusion matrix for test tile (excluding Non-Forest class). OA, weighted mean F1 = 87%, Kappa = 74%. Diagonal elements and totals in bold.

Pixel Count	Label Data									User’s
RF	Acacia	Casuarina	Eucalyptus	Grassland	Mangrove	Melaleuca	Plantation	Rainforest	Total	acc. %
Acacia	**150**	15	2209	24,962	0	0	488	71	**27,895**	0.5
Casuarina	61	**386**	109,680	710	0	0	3535	83	**114,455**	0.3
Eucalyptus	36,146	67,505	**9,267,052**	642,501	2489	1545	392,928	108,411	**10,518,577**	88.1
Grassland	2168	718	86,471	**3,656,064**	73	100	60,653	75	**3,806,322**	96.1
Mangrove	1394	790	41,403	5456	**3197**	339	3165	182	**55,926**	5.7
Melaleuca	362	238	13,231	264	4	**11**	477	141	**14,728**	0.1
Plantation	751	182	151,468	158,140	13	5	**348,107**	11,212	**669,878**	52.0
Rainforest	85	38	14,263	94	0	0	1209	**3515**	**19,204**	18.3
Total	**41,117**	**69,872**	**9,685,777**	**4,488,191**	**5776**	**2000**	**810,562**	**123,690**	**15,226,985**	
Prod.‘s acc. %	0.4	0.6	95.7	81.5	55.3	0.6	42.9	2.8		
F1 %	0.4	0.4	91.7	88.2	10.4	0.1	47.0	4.9		

**Table 6 jimaging-10-00143-t006:** Predicted vegetation classes for 2000 and 2019, showing changes.

Class	2000 (ha)	2019 (ha)	Change (ha)	Percent
Non–forest	35,545	34,966	−580	−1.6%
Acacia	751	754	3	0.3%
Casuarina	6604	5541	−1063	−16.1%
Eucalyptus	611,136	619,204	8068	1.3%
Grassland	301,425	283,423	−18,002	−6.0%
Mangrove	528	358	−170	−32.2%
Melaleuca	589	415	−174	−29.5%
Plantation	33,252	45,828	12,576	37.8%
Rainforest	10,170	9511	−660	−6.5%
Total	1,000,000	1,000,000		

## Data Availability

The data presented in this study are available on request from the corresponding author.

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
