# Peer review of "U-Net Convolutional Neural Network for Mapping Natural Vegetation and Forest Types from Landsat Imagery in Southeastern Australia"

_2313-433X, 2024, doi:10.3390/jimaging10060143_

Round 1

Reviewer 1 Report

Comments and Suggestions for Authors

The paper titled "Convolutional Neural Network outperforms Random Forests in mapping natural vegetation and forest types from Landsat imagery in southeastern Australia" aims to compare the performance of Convolutional Neural Networks (CNNs) and Random Forests (RF) in mapping natural vegetation and forest types in southeastern Australia using Landsat imagery. The manuscript is well structured and provides a clear overview of the research objectives, methodology, results and discussion. The manuscript seems to be in line with the aims and scope of the Journal of Imaging, but I suggest some points to be revised.  The authors should address the points mentioned below to improve the clarity, soundness, and impact of the study.

Line 167: The methodology section lacks detailed information on certain aspects, such as the specific parameters and hyperparameters used in the CNN and RF models. Please provide detailed information.

Line 191: Please provide more details about the data augmentation process to understand how the models are trained.

Line 285: Presents the performance metrics of the CNN and RF models, a more in-depth analysis and interpretation of the findings would be beneficial. Explaining the implications of the results in the context of vegetation mapping and highlighting the significance of the differences between the models would add value to the manuscript.

Line 386: Discussing the limitations of the study, potential sources of error, and avenues for future research would enhance the discussion.

Reviewer 2 Report

Comments and Suggestions for Authors

In this work, a U-Net CNN is used to map natural vegetation and different types of forests from satellite images of southeastern Australia. The results are mainly compared with the random forests method. Overall, the work is interesting; however, several aspects need to be addressed to demonstrate contribution and applicability.

Formatting comments:

1)     The font size in several figures is not legible. The authors should ensure that the font size in the figures matches the font size of the article text.

Content comments:

1)     Line 67 already states that deep learning schemes yield better results than RF. Why study this comparison then?

2)     It is not clear why RF is being compared. The authors need to justify this.

3)     A comparison should be made with the methods reported in the literature, whether machine learning or deep learning methods.

4)     It is not clear why a U-Net CNN is used. There are many more CNN proposals in the literature. The authors should compare them and, based on the results, select the best one.

5)     Based on comment 1, objectives 1 and 3 seem to lack contribution.

6)     Examples of images of the classes presented in Table 1 would help to better understand the differences.

7)     Since U-Net CNN and the RF method already exist, it is unclear what the contribution is. If they are just used and the results are discussed, the work lacks contribution.

8)     How were the percentages shown in Table 2 selected? Do the results change if other percentages are used? Was k-fold used for validation? If not, these results should be included and discussed.

9)     Again, lines 188 and 199 show parameters selected from other works, so what is the contribution? What happens if other parameters are used? This should be clearly discussed in the article.

10)  What is the impact on the compared methods when the class data is unbalanced? Was any data augmentation strategy used?

11)  For the CNN, the accuracy and loss graphs should be shown and analyzed.

12)  The software and hardware characteristics used should be mentioned.

13)  A comparative table of advantages, disadvantages, and limitations should be included. A table comparing the obtained results with those reported in the literature in quantitative and qualitative terms should be added to make the contribution of the work evident.

14)  The computational cost should be discussed.

Comments on the Quality of English Language

Minor editing of English language required

Reviewer 3 Report

Comments and Suggestions for Authors

The title is not agile and clear. How about changing it to "U-Net for mapping natural vegetation and forest types from Landsat imagery in southeastern Australia". TBH, the first moment I saw the original title I was surprised a regular CNN can outperform RFs on these types of tasks. However, the authors are actually using U-Net, which completely makes sense. I'd suggest the authors to be more specific on what models are used in this paper. I have some additional comments.

1. How was the Random Forest model set up in terms of tree depth and number of estimators? What considerations guided these choices?

2. The authors mention evaluating the temporal stability of the CNN model outputs. Could you detail the methodology used for this analysis? How did you define and measure temporal stability in the context of annual vegetation mapping?

3. How were the six Landsat bands chosen, and were there considerations for including or excluding additional bands based on their relevance to vegetation type classification?

Round 2

Reviewer 2 Report

Comments and Suggestions for Authors

Comments and suggestions have been addressed. 

Comments on the Quality of English Language

Minor editing of English language required

Reviewer 3 Report

Comments and Suggestions for Authors

The revised manuscript looks good to me. I don't have more comments.